# DFL$^2$G: Dynamic Agnostic Federated Learning with Learngene

## Abstract

Dynamic agnostic federated learning is a promising research field where agnostic clients can join the federated system at any time to collaboratively construct machine learning models. The critical challenge is to securely and effectively initializing the models for these agnostic clients, as well as the communication overhead with the server when participating in the training process. Recent research usually utilizes optimized global model for initialization, which can lead to privacy leakage of the training data. To overcome these challenges, inspired by the recently proposed Learngene paradigm, which involves compressing a large-scale ancestral model into meta-information pieces that can initialize various descendant task models, we propose a **D**ynamic agnostic **F**ederated **L**earning with **L**earn**G**ene framework. The local model achieves smooth updates based on the Fisher information matrix and accumulates general inheritable knowledge through collaborative training. We employ sensitivity analysis of task model gradients to locate meta-information (referred to as *learngene*) within the model, ensuring robustness across various tasks. Subsequently, these well-trained *learngenes* are inherited by various agnostic clients for model initialization and interaction with the server. Comprehensive experiments demonstrate the effectiveness of the proposed approach in achieving low-cost communication, robust privacy protection, and effective initialization of models for agnostic clients.

## 1 Introduction

Federated Learning (FL) (McMahan et al., 2017) has shown great promise in the field of distributed learning across devices, allowing multiple clients to collaboratively train a shared global model without exposing private data (Chen et al., 2022). Each client trains a local model based on its private data and then shares its high-dimensional model parameters with the server for collaborative learning across devices in FL. Recently, the integration of FL has improved the security and efficiency of practical applications in the field of artificial intelligence, including medical impact analysis (Ng et al., 2021; Rieke et al., 2020; Guan et al., 2024; Jiang et al., 2022), personalized recommendation system (Wu et al., 2023; Imran et al., 2023) and intelligent transport system (Shinde & Tarchi, 2023; Pandya et al., 2023).

In real-world FL scenarios, it is crucial to maintain the privacy goals of FL while reducing costs to improve system efficiency (Lyu et al., 2020; Niknam et al., 2020). Recent research has proposed advanced methods such as model pruning compression (Karimireddy et al., 2020; Haddadpour et al., 2021), one-shot FL (Jhunjhunwala et al., 2024; Elmahallawy & Luo, 2023; Zhang et al., 2022; Andrew et al., 2024), and reducing local updates to achieve controllable communication costs (Karimireddy et al., 2020). Correspondingly, Dynamic Agnostic Federated Learning (DAFL), which involves the agnostic clients continuously join into the FL system for model training, also contains low communication costs and high privacy two fundamental goals. Moreover, effectively initializing these models to achieve stable convergence is a significant challenge. Generally, the use of pre-trained global model parameters for initialization inevitably exposes privacy risks, and leads to overfitting to the trained data while inadequately adapting to agnostic data distributions (Zhu et al., 2019; Nguyen et al., 2022). This highlights the core objective of DAFL: *How to design a scheme that enables efficient and secure communication between clients and the server, while ensuring effective initialization of agnostic models?*

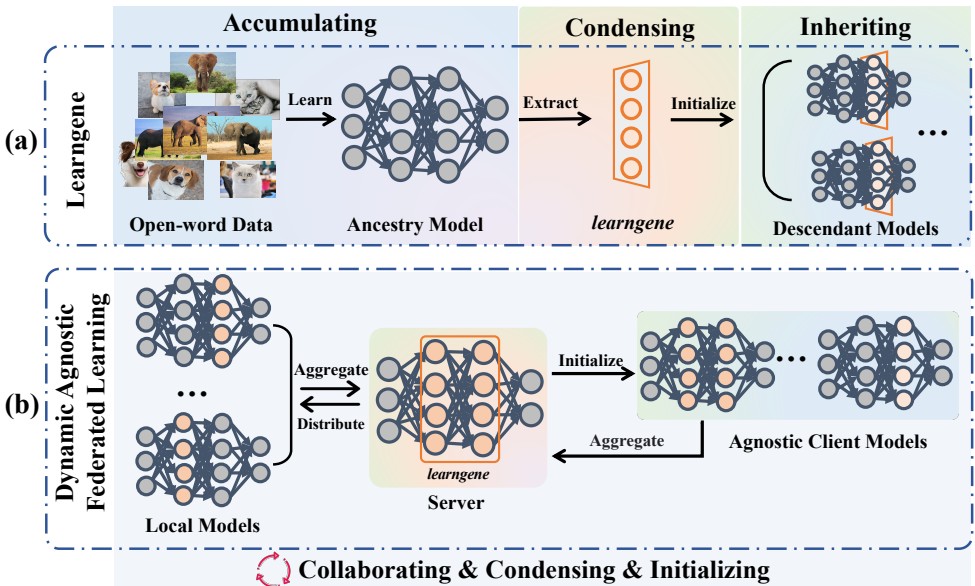

Figure 1: Illustration of Dynamic Agnostic Federated Learning and Learngene. In the accumulating, condensing and inheriting processes of the Learngene, dynamic agnostic federated learning can achieve a corresponding organic integration.

To achieve this goal, we were inspired by a novel and practical machine learning paradigm, Learngene (Wang et al., 2022; 2023). It is based on mechanisms from biological genetics, condensing a large-scale ancestral model into lightweight genes, which are then inherited by descendant task models in various scenarios. Specifically shown in Figure 1 (a), large-scale ancestral model training learns from open-world data to **accumulate** knowledge, and **condenses** it to obtain a lightweight information piece (i.e., *learngene* [1]) with high controllability, privacy, and low deployment costs, enabling various descendant models to **inherit** these *learngenes* for rapid and effective initialization.

We attempt to effectively integrate the Learngene paradigm with the Dynamic Agnostic Federated Learning scenes, as illustrated in Figure 1 (b) : **(i) Collaborating:** the local models are smoothly updated and collaboratively trained to accumulate knowledge; **(ii) Condensing:** the local models are condensed into lightweight *learngenes* for interaction with the server, and encapsulated in the global model, which is then stored in the server; **(iii) Initializing:** the *learngenes* are used to rapidly and efficiently initialize agnostic models, which are then participate in the collaborative training. These three processes can be parallelized in DAFL. In addition, one form of *learngene* expression is configured to retain multiple complete layers (Wang et al., 2023). For various tasks, satisfactory performance can be achieved with a few number of samples by inheriting the *learngene* to initialize descendant models.

With this in mind, we propose a **D**ynamic agnostic **F**ederated **L**earning with **L**earn**G**ene (DFL$^2$G) framework, which consists of three modules: Learngene Smooth Learning, Learngene Dynamic Aggregation, and Learngene Initial Agnostic Model. To mitigate the issue of communication time per round in typical FL, which is influenced by the slowest participating client when using a single global model, we introduce the one-shot clustering method to obtain multiple cluster models. Furthermore, the local models updating within each cluster are rely on the respective cluster model to assimilate knowledge from other participants, and use layer-wise Fisher information values to partition the elastic *learngene* for quadratic regularization. Finally, participating models obtain their individual *learngene* based on the similarity metric with the historical local model, which is then uploaded to the server for aggregation to obtain cluster *learngene* for subsequent model updates or agnostic model initialization. DFL$^2$G can seek for *learngene* during the collaborative learning process of existing local models, which can reduce communication costs and facilitate the initialization of dynamically

---

[1]"Learngene" refers to the learning framework, while "*learngene*" denotes the condensed information piece of the model. For detailed background information, please refer to the "Related Work" section in the Appendix A.1.

participating agnostic client models to achieve stable performance improvements. In summary, our main contributions are summarized as follows:

- We propose the "Collaborating & Condensing & Initializing" mechanism in dynamic federated learning, inspired by the "Accumulating & Condensing & Inheriting" of the Learngene paradigm to improve model interpretability.

- We propose a dynamic agnostic federated learning with Learngene framework, which seeks *learngene* to safely and cheaply interact between the clients and server during model optimization and to efficiently initialize agnostic client models.

- Extensive experiments demonstrate DFL$^2$G's competitive performance in both agnostic clients initialization and communication costs reduction, with a reduction of approximately **9.2** $\times$ parameters compared to FEDAVG. Furthermore, privacy analysis confirms DFL$^2$G's robust privacy protection against adversarial gradient inversion attacks.

## 2 METHODOLOGY

### 2.1 PROBLEM FORMULATION

In practical FL applications, there is heterogeneity among clients and some unknown new clients may join the FL system at any time. Let $\mathcal{N}$ be the set of known clients with the size of $N$, where the non-iid distributed training data on $i$-th client is denoted as $\mathcal{D}_i = \{(x_i, y_i)\}$, $i \in \mathcal{N}$, $x_i, y_i$ are the corresponding data pair. Similarly, $\mathcal{M}$ denotes the set of agnostic clients with the size of $M$. The class sets of the agnostic clients $\mathcal{C}_j$ for $j \in \mathcal{M}$ are disjoint from the class sets of the known clients $\mathcal{C}_i$ for $i \in \mathcal{N}$, expressed as $\mathcal{C}_j \cap \left( \bigcup_{i \in \mathcal{N}} \mathcal{C}_i \right) = \emptyset$.

Additionally, we aim to group clients with similar data distributions, such that clients within the same cluster can leverage each other's data for improved performance in federated learning. On the server side, the known clients $\mathcal{N}$ that have already participated in training are grouped into $K$ clusters (denoted as $k$) based on the distributional similarity between their data subspaces, using a one-shot clustering approach as detailed in (Vahidian et al., 2023). Therefore, the server contains $K$ cluster models, where a client $i$ belonging to cluster $k$ has a parameterized classification network $\theta_{k,i}$, and the corresponding cluster model is $\Theta_k$. Generally, each client $i$ optimizes its model by minimizing the classification loss, as follows:

$$\mathcal{L}_{cls} = \mathbb{E}_{x_i, y_i \sim \mathcal{D}_i} \Phi \left( (x_i \mid y_i; \theta_i), y_i \right), \tag{1}$$

where $\Phi$ is the Cross-Entropy loss function and $y_i$ is the ground truth label.

### 2.2 METHOD OVERVIEW

The proposed **D**ynamic agnostic **F**ederated **L**earning with **L**earn**G**ene (DFL$^2$G) framework consists of three modules: Learngene Smooth Learning, Learngene Dynamic Aggregation and Learngene Initial Agnostic Model. An illustration of the learning procedure is shown in Figure 2. During the $t$-th epoch, the local models perform smooth updates based on the cluster model and execute quadratic regularization (Sun et al., 2023) using the elastic *learngene* partitioned by the Fisher Information Matrix (FIM) to improve the adaptability of the local models to the client data distribution. The optimal *learngene* is identified based on the layer similarity score $\xi$ with the previous model $\tilde{\theta}_{k,i}$. Participating clients then upload their individual *learngene* $(\theta_{\mathcal{G}_{k,1}}, \cdots, \theta_{\mathcal{G}_{k,i}})$ to the server for dynamic aggregation and subsequent distribution to them. When the agnostic client makes a model request, the server sends the nearest cluster *learngene* to facilitate the initialization of its model.

### 2.3 LEARNGENE SMOOTH LEARNING

During local updates, the goal is to compress a unique *learngene* for each client, thereby reducing communication costs interacting with the server. In particular, local client models approximate the corresponding cluster model constraints for smooth updates and apply quadratic regularization on the elastic *learngene* partitioned by FIM to effectively capture generalizable knowledge.

**Smooth Updating Based Cluster Model.** We leverage the cluster model obtained after collaborative learning to impose smooth constraints on the local models. These models trained locally exhibit

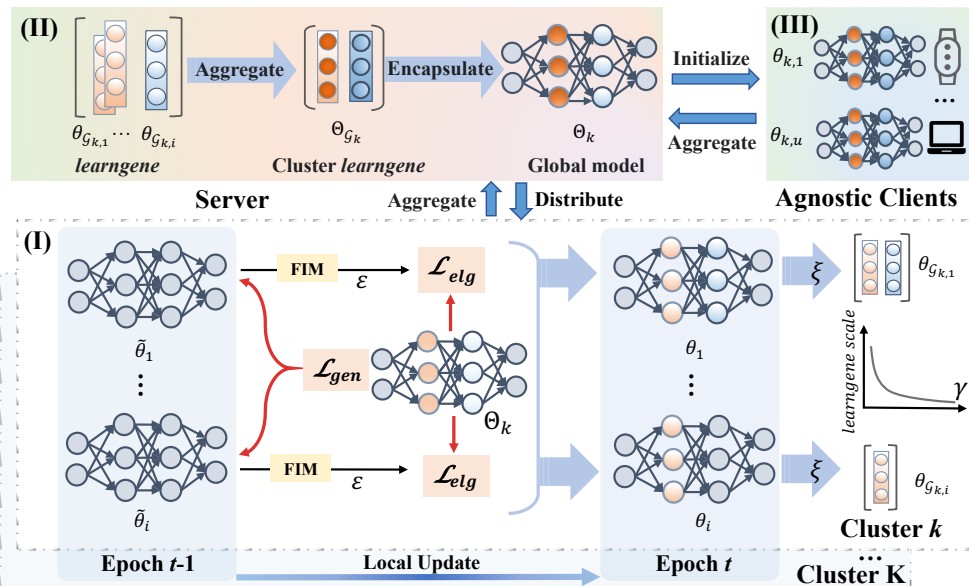

Figure 2: Illustration of training process of DFL$^2$G, which includes (I) Learngene Smooth Updating, (II) Learngene Dynamic Aggregation, and (III) Learngene Initial Agnostic Model.

similarity within the same cluster, uploading their itself lightweight *learngene* for client-server interaction, effectively mitigating communication costs. Traditional federated learning methods typically apply a uniform regularization strength to all parameters of the model, ignoring inherent differences in magnitudes among parameters, leading to biased selection of the *learngene*. To address this, we propose an adaptive smoothness constraint based on the each layer, taking into account the magnitude of its parameters to determine the strength of the constraint. The local update of the $i$-th model is expressed as:

$$\theta_{k,i} \leftarrow \tilde{\theta}_{k,i} - \alpha \nabla_\theta \mathcal{L}_{gen}(\tilde{\theta}_{k,i}, \mathcal{D}_i), \tag{2}$$

where $\mathcal{D}_i$ represents the private dataset of client $i$ within $k$-th cluster, $\alpha$ is the learning rate, $\tilde{\theta}_{k,i}$ is the previous model, and $\nabla_\theta \mathcal{L}_{gen}$ is the gradient of the first smooth loss function. This is calculated as the sum of the Cross-Entropy loss and the L2 norm of the difference between the current model parameters $\theta_{k,i}$ and the cluster model parameters $\Theta_k$:

$$\mathcal{L}_{gen} = \|\theta_{k,i} - \Theta_k\|_2. \tag{3}$$

**Smooth Updating Based Elastic *Learngene*.** Considering the effective initialization of agnostic client models during the learning process, lightweight *learngene* should have high informational content to effectively predict untrained classes. Fisher information matrix (FIM) quantification of model parameters provides rich information content (Jhunjhunwala et al., 2024; 2023; Yan et al., 2022; Shoham et al., 2019; Yang et al., 2023). Inspired this, we introduce the FIM that diagonal is used to weight the importance of each parameter of the model and determine the penalty size for changing the parameter in client training. We can obtain a good approximation to the diagonal of the Fisher values for each parameter indexed by $j$ in the model $\tilde{\theta}_i$ (refers to $\tilde{\theta}_{k,i}$), as follows:

$$F_{i,j} = \mathbb{E}\left[\left(\frac{\partial \log h(\tilde{\theta}_i \mid \mathcal{D}_i)}{\partial \tilde{\theta}_{i,j}}\right)^2\right], \tag{4}$$

where $h(\tilde{\theta}_i \mid \mathcal{D}_i)$ is the likelihood function that represents the fitness of the model parameters under the data $\mathcal{D}_i$. The elastic *learngene* we seek is a model fragment that reflects the shared knowledge of the sample data, selected based on Fisher values computed from the model and private data. Specifically, if the Fisher value is below a given threshold $\varepsilon$, the model parameters with small local model update variations (i.e., its own *learngene*) are retained, while the remaining parameters are updated using the aggregated consensus model (knowledge shared across clients):

$$\theta'_{k,i} = \begin{cases} \tilde{\theta}_{k,i,j}, & \hat{F}_{i,j} \leq \varepsilon \\ \Theta_{k,j}, & otherwise, \end{cases} \tag{5}$$

where $\hat{F}_{i,j} = \frac{F_{i,j} - \min(F_i)}{\max(F_i) - \min(F_i)}$ represents the normalization of $F_{i,j}$.

The local model is restructured according to the FIM into a combination of the *learngene* and the cluster model, aiming to approximate its own *learngene* to the cluster elastic *learngene*. This enables learning of shared knowledge within the same cluster to reduce redundant parameters, thereby lowering communication costs. Then the quadratic regularization update of the current local model based on the cluster elasticity *learngene* is expressed as:

$$\theta_{k,i} \leftarrow \theta_{k,i} - \alpha \nabla_\theta \mathcal{L}_{elg}(\theta'_{k,i}, \mathcal{D}_i), \tag{6}$$

where $\nabla \mathcal{L}_{elg}$ is the gradient of the elasticity loss $\mathcal{L}_{elg}$, calculated as the L2 norm of the difference between the local model's *learngene* and the cluster's elastic *learngene*:

$$\mathcal{L}_{elg} = \left\| \theta'_{k,i} - \Theta_k \right\|_2. \tag{7}$$

In summary, the total optimization objective for local updates is defined as $\mathcal{L}_{all} = \mathcal{L}_{cls} + \lambda_1 \mathcal{L}_{gen} + \lambda_2 \mathcal{L}_{elg}$, where $\mathcal{L}_{cls}$ represents the classification loss, while $\lambda_1$ and $\lambda_2$ serve as hyperparameters that regulate $\mathcal{L}_{gen}$ and $\mathcal{L}_{elg}$, respectively.

**Localization of *Learngene*.** After training the local model based on private data, we precisely identify each client's *learngene*, enabling the acquisition of meta-knowledge about the model. The strategy is determining the contribution of each layer, guided by the parameter changes observed after model training. The score of the $l$-th layer $\xi_{k,i}^l$ can be calculated on the locally trained model $\theta_i$ and previous model $\tilde{\theta}_i$, as follows:

$$\xi_{k,i}^{(l)} = \frac{cos\left(\theta_{k,i}^{(l)}, \tilde{\theta}_{k,i}^{(l)}\right)}{dim\left(\theta_{k,i}^{(l)}\right)}, \tag{8}$$

where $dim(\cdot)$ denotes the number of parameters on layer $l$, which can normalize the values as $\sum_{l=1}^{L} \xi_{k,i}^{(l)} = 1$. $cos$ is the cosine measure which can take a variety of forms, such as L1, L2, and Earth Mover distance. Here, $\xi_{k,i}^l$ quantifies the discrepancy in the $l$-th layer between $\theta_i$ and $\tilde{\theta}_{i,l}$, thereby evaluate the personalized influences on the $l$-th layer of the current model. Intuitively, a higher $\xi_{k,i}^l$ value suggests a greater deviation of the $l$-th layer in $\theta_i$ from $\tilde{\theta}_{i,l}$, indicating a more pronounced impact on personalization. Conversely, lower $\xi_{k,i}^l$ values indicate a higher contribution to generalization information, which is beneficial for initializing new tasks, which is exactly what we seek *learngene*. The symbol $S_{k,i}$ is the $\xi_{k,i}^{(l)}$ values calculated for the $L$ layers, arranged in descending order, setting the round $\gamma$ layer to 1 and the others to 0, and then obtaining the updated $\theta_{\mathcal{G}_{k,i}^{(l)}} = \theta_{k,i} \odot S_{k,i}$. The *learngene* progressively tightens throughout the training and update process until it reaches a threshold layer $\gamma$, which is determined by a performance-based adaptive training process. This parallel procedure, involving both model updates and the localization of the *learngene*, enables the model to fit the data distribution while achieving the goal of reducing communication costs.

## 2.4 LEARNGENE DYNAMIC AGGREGATION

In the server, our goal is to maintain a unified cluster *learngene* for each cluster, which encapsulate the generalization parameter information of all relevant local models within the cluster, allowing effective initialization of newly agnostic client models. The *learngene* layers that are common to all participants are aggregated to obtain clusters *learngene*, while the others retain the previous cluster model. The formula for aggregating cluster *learngene* is as follows:

$$\Theta_{\mathcal{G}_k^{(l)}} = \frac{1}{N_k^{(l)}} \sum_{i=1}^{N_k^{(l)}} \theta_{\mathcal{G}_{k,i}^{(l)}}, \tag{9}$$

where $l \in L$ and $\Theta_{\mathcal{G}_k^{(l)}}$ represents the parameters of the $l$-th layer within the $k$-th cluster *learngene*. Additionally, $N_k^{(l)}$ denotes the number of client *learngenes* that encompass the $l$-th layer, while $\theta_{\mathcal{G}_{k,i}^{(l)}}$ signifies the parameters of the $l$-th layer within the $i$-th *learngene* belonging to the $k$-th cluster. The updated cluster model is then represented as the aggregated *learngene* and the previous partial cluster model parameters: $\Theta_k = [\Theta_{\mathcal{G}_k}; \tilde{\Theta}_k]$.

## 2.5 LEARNGENE INITIAL AGNOSTIC MODEL

In the dynamic and agnostic FL scenario, when a new client $i$ joins, we recommend applying truncated singular value decomposition (Li & Xie, 2024) to its private data sample $\mathcal{X}_i$ to obtain the components that describe the underlying data distribution. Specifically, we define the decomposition as:

$$\mathcal{X}_{i,d} = \mathbf{U}_{i,d}\boldsymbol{\Sigma}_{i,d}\mathbf{V}_{i,d}^T, \tag{10}$$

where $\mathbf{U}_{i,d} = [\mathbf{u}_1, \mathbf{u}_2, \ldots, \mathbf{u}_d] \in \mathbb{R}^{m \times d}$ (with $d \ll \text{rank}(\mathcal{X}_i)$ and $m$ denotes the number of samples for client $i$) represents the top $d$ most significant left singular vectors, capturing the essential features of the underlying data distribution. We follow the (Vahidian et al., 2023) and select $d = 5$ to mitigate the risk of data leakage. Additionally, to facilitate linear algebraic computations, we transform the matrix $\mathbf{U}_{i,d}$ into a vector $\mathbf{u}_{i,d} \in \mathbb{R}^{md \times 1}$.

Then, client $i$ upload $\mathbf{u}_{i,d}$ to the server for requesting *learngene*. Let $\mathbf{u}_k$ be the mean vector of the $k$-th cluster. The server calculates the distance $d_{i,k}$ between $\mathbf{u}_{i,d}$ and $\mathbf{u}_k$ as follows:

$$d_{i,k} = \|\mathbf{u}_{i,d} - \mathbf{u}_k\|. \tag{11}$$

The server identifies the nearest cluster $k$ based on these calculated distances and transmits the associated *learngene* $\Theta_{\mathcal{G}_k}$ from that cluster to the requested agnostic client for model initialization. For the agnostic client model $\theta_{k,i}$, the initialization parameters consist of two components: inherited cluster *learngene* $\Theta_{\mathcal{G}_k}$ and random initialization $\theta_0$, expressed as $\theta_{k,i} = [\theta_0; \Theta_{\mathcal{G}_k}]$.

## 2.6 PRIVACY ANALYSIS

Typically, the initialization of agnostic client models benefits from the server-side model, while collaborative learning among clients necessitates communication with the server. Therefore, the proposed method should emphasize the importance of privacy guarantees on the server side. In the validation phase of this study, we treat the server as a malicious entity capable of reconstructing the original data from a victim client using the iDLG method (Zhu et al., 2019; Wu et al., 2021). The $\mathcal{L}_D$ loss associated with recovering the true data from the victim client $i$ is calculated as follows:

$$\mathcal{L}_D = \|\nabla_\Theta \mathcal{L}_{cls}(x_i) - \nabla_\Theta \mathcal{L}_{cls}(\tilde{x})\|^2, \tag{12}$$

where $x_i$ is the real data of victim client $i$ while $\tilde{x}$ is the variable to be trained to approximate $x_i$ by minimizing $\mathcal{L}_D$ that is the distance between $\nabla_\Theta \mathcal{L}_{cls}(x_i)$ and $\nabla_\Theta \mathcal{L}_{cls}(\tilde{x})$. The former is observed gradients of $\mathcal{L}_{cls}$ (see Eq. (1)) w.r.t. model parameters $\Theta$ for the real data $x_i$, while the latter is estimated gradients for $\tilde{x}$. We evaluated the privacy guarantees of the DFL$^2$G, FEDAVG (McMahan et al., 2017), and PartialFed (Sun et al., 2021) methods. For the FEDAVG, which shares the entire network, we set $\Theta := \theta_i$. For PartialFed, where only selected network layers are uploaded to the server, $\Theta := [\theta_0; \theta_s]$ that $\theta_0$ is the random initialization parameter. Similarly, in DFL$^2$G, only the *learngene* is shared, so $\Theta := [\theta_0; \theta_{\mathcal{G}_i}]$. Since the network used for training is ResNet model, we employ the same network for validation, with MSE utilized as the loss function to evaluate the quality of the image reconstruction.

## 2.7 DISCUSSION

We describe the whole training process of DFL$^2$G is shown in Algorithm 1, which includes local update, server execution and agnostic client model initialization processes. The corresponding describes the local model update and the selection of *learngenes*, the dynamic aggregation of *learngenes* in the server , and the initialization of agnostic client models. The optimization process primarily focuses primarily on the participants, interacting with the server using a small-scale *learngene*, and the server generally aggregating *learngene* and responding to agnostic clients.

Our analysis of the additional computational cost is as follows: Suppose each global epoch consists of $E_l$ local update epochs, $N$ clients, and each model contains $P$ parameters. Since the local model needs to compute Fisher information values, it incurs an additional computational cost of $\mathcal{O}(P)$. Therefore, each global round introduces a computational cost of $\mathcal{O}(E_l \cdot N \cdot P)$.

---

**Algorithm 1:** DFL$^2$G

---

**Input:** Local epochs $E_l$, participants number in the $k$-th cluster $N_k$, private data of the $i$-th
praticipant $\mathcal{D}_i$, global model parameters $\Theta_k$, local model parameters $\theta_{k,i}$ and previous
local model parameters $\tilde{\theta}_{k,i}$, cluster *learngene* $\Theta_{\mathcal{G}_k}$ of the cluster $k$ and local *learngene*
$\theta_{\mathcal{G}_{k,i}}$, hyper-parameter $\lambda, \varepsilon, \gamma$, learning rate $\alpha$;

1    **Local Update :**
2    **for** $i = 1, 2, \cdots, N_k$ *in parallel* **do**
3       Receive $\Theta_k$ from server;
4       $F_{k,i} \leftarrow (\mathcal{D}_i, \tilde{\theta}_{k,i})$ by Eq. (4);
5       **for** $e = 1, 2, \cdots, E_l$ **do**
6         $\theta_{k,i} \leftarrow \tilde{\theta}_{k,i} - \alpha \nabla_\theta \mathcal{L}_{gen}(\tilde{\theta}_{k,i}, \mathcal{D}_i)$ by $\mathcal{L}_{gen}$ from Eq. (3);
7         $\theta'_{k,i} \leftarrow (F_{k,i}, \Theta_k, \tilde{\theta}_{k,i}, \varepsilon)$ using Eq. (5);
8         $\theta_{k,i} \leftarrow \theta_{k,i} - \alpha \nabla_\theta \mathcal{L}_{elg}(\theta'_{k,i}, \mathcal{D}_i)$ by $\mathcal{L}_{elg}$ from Eq. (7);
9       **end**
10      $\xi_{k,i} \leftarrow (\tilde{\theta}_{k,i}, \theta_{k,i})$ using Eq. (8);
11      $\theta_{\mathcal{G}_{k,i}} \leftarrow \theta_{k,i} \odot S_{k,i}$ by sort the $\xi_{k,i}$ of the $L$ layers to obtain the mask set $S_{k,i}$ of the
corresponding layer;
12    **end**
13    **Server Execute :**
14    $\theta_{\mathcal{G}_{k,i}} \leftarrow$ **Local Update** $(\Theta_k)$ ;
15    $\Theta_{\mathcal{G}_k^{(l)}} \leftarrow \frac{1}{N_k^{(l)}} \sum_{i=1}^{N_k^{(l)}} \theta_{\mathcal{G}_{k,i}^{(l)}}$ from the Eq. (9) then $\Theta_k \leftarrow [\Theta_{\mathcal{G}_k}; \tilde{\Theta}_k]$;
16    Send $\Theta_k$ to the participate in training clients;
17    **if** *Agnostic client request* **then**
18      Select the nearest cluster $k$ by Eq. (11);
19      Send the cluster *learngene* $\Theta_{\mathcal{G}_k}$;
20      $\mathbf{u}_k \leftarrow \frac{N_k \cdot \mathbf{u}_k + \mathbf{u}_{i,d}}{N_k + 1}$, where $N_k$ denotes the number of clients in cluster $k$;
21    **end**
22    **Agnostic Client Initialize :**
23    Send the $\mathbf{u}_{i,d}$ by Eq. (10) to server;
24    Receive $\Theta_{\mathcal{G}_k}$ from server;
25    $\theta_{k,i} \leftarrow [\theta_0; \Theta_{\mathcal{G}_k}]$;

---

# 3 EXPERIMENTS

## 3.1 EXPERIMENTAL SETUP

### 3.1.1 DATASETS AND DATA PARTITION

Our experiments are conducted on the following three real-world datasets: SVHN (Netzer et al., 2011), CIFAR10 (Krizhevsky et al., 2009), and CIFAR100 (Krizhevsky et al., 2009). To simulate real-world applications, we employ a classic *Sharding* strategy to generate non-iid data partitions among clients, where $s$ denotes the number of classes contained in each client, constrained not to exceed the total number of classes. By varying the parameter $s$, we obtain different non-iid data distributions. For SVHN and CIFAR10, we select $s = \{4, 5\}$, while for CIFAR100 we set for $s = \{10, 20\}$. Agnostic clients and seen clients are sampled from the same dataset, with completely different classes and no overlap between samples.

### 3.1.2 BASELINES

To ensure a fair comparison, we selected six baseline FL methods, including FEDAVG (McMahan et al., 2017), which involves the interaction of all model parameters between server and clients; PartialFed (Sun et al., 2021), which initializes a subset of global model parameters; FedFina, which incorporates rich information in the last four layers of the model; FedLPS (Jia et al., 2024) and FedLP (Zhu et al., 2023), which use model pruning to compress and reduce communication costs in

federated learning algorithms. Flearngene (Wang et al., 2023), a lightweight *learngene* presented for the first time, extracts information from gradient updates. All methods were implemented under the same data distribution as well as clustering process and device environments.

### 3.1.3 SETTINGS

For local client training, we employ the ResNet18 model (He et al., 2016) to perform classification tasks. The stochastic gradient descent optimizer is used with a momentum of 0.9 and a learning rate of 0.01 is utilized. We set the batch size to 64, the number of epochs for global collaborative accumulation training to 100, the number of local epochs to 10, and the number of subsequent training epochs for the initialization-agnostic client model to 50. The specific hyperparameters are described in Appendix A.2.2.

### 3.2 EXPERIMENT RESULTS

We conducted a comprehensive evaluation of the proposed method, covering three main aspects: communication costs, testing performance of agnostic models, and privacy protection within dynamic federated learning scenarios. Furthermore, ablation analysis was performed on various loss functions to ascertain the significance and necessity of each component.

**Low-cost Communication Evaluation.** The communication costs typically involves a cycle of data upload from the client to the server and subsequent download from the server to clients. Since experiments typically involve downloading aggregated global model, we discuss the communication costs of clients uploading parameters to the server. We propose a cost-effectiveness metric ($Cef = \frac{Comm}{Acc}$), which is the ratio of communication cost (*Comm*, GB) to model performance (*Acc*, %). This metric allows for a relatively fair evaluation of the performance of federated model pruning methods.

Table 1: Comparison with state-of-the-art methods on *Comm* (↓) and *Cef* (↓) metrics during the accumulation process.

| Methods | SVHN | | | | CIFAR10 | | | | CIFAR100 | | | |
|---|---|---|---|---|---|---|---|---|---|---|---|---|
| | s = 4 | | s = 5 | | s = 4 | | s = 5 | | s = 10 | | s = 20 | |
| | *Comm* | *Cef* | *Comm* | *Cef* | *Comm* | *Cef* | *Comm* | *Cef* | *Comm* | *Cef* | *Comm* | *Cef* |
| FEDAVG | 15.41 | 0.1675 | 14.21 | 0.1580 | 12.04 | 0.1488 | 15.00 | 0.1885 | 13.30 | 0.2574 | 13.39 | 0.3590 |
| PartialFed | 4.32 | 0.0507 | 3.98 | 0.0488 | 3.37 | 0.0576 | 4.20 | 0.0633 | 4.33 | 0.0869 | 3.74 | 0.1026 |
| FedFina | 11.38 | 0.1459 | 10.49 | 0.1403 | 8.89 | 0.1347 | 11.07 | 0.1814 | 11.41 | 0.2574 | 9.84 | 0.3165 |
| FedLP | 11.65 | 0.1264 | 10.93 | 0.1215 | 9.41 | 0.1161 | 12.24 | 0.1575 | 11.90 | 0.2298 | 10.89 | 0.2985 |
| FedLPS | 3.34 | 0.0377 | 3.93 | 0.0452 | 4.00 | 0.0527 | 3.49 | 0.0512 | 3.12 | 0.1698 | 3.21 | 0.1550 |
| Flearngene | 6.61 | 0.0789 | 6.09 | 0.0776 | 5.16 | 0.0734 | 6.43 | 0.0990 | 5.68 | 0.1249 | 5.71 | 0.1870 |
| Ours | 2.74 | **0.0323** | 2.62 | **0.0321** | 3.54 | **0.0525** | 4.28 | 0.0571 | 3.08 | **0.0712** | 3.02 | **0.1011** |

Table 1 presents a comparison of the average communication costs and cost-effectiveness over all training epochs for different methods on various datasets. Note that we highlight the **Best** results in bold and the Second-best results are underlined. Compared to FEDAVG, the proposed method shows a significant reduction in communication costs and an increased cost-effectiveness. Compared to the pruning method FedLPS, it reduces communication overhead by 0.66 GB while demonstrating lower cost-effectiveness on the CIFAR100 dataset. This demonstrates that our method achieves higher model accuracy with lower communication costs, making it more efficient in terms of resource utilization and well-suited for FL systems facing communication bottlenecks.

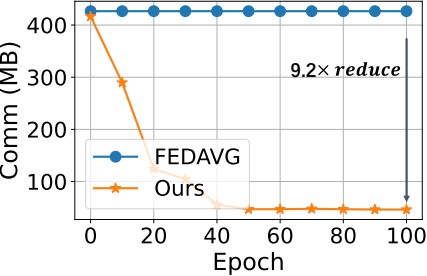

Figure 3: Communication cost curves.

Table 2: Required *Comm* (GB) for target *Acc* (%).

| Methods | SVHN s=5 | CIFAR10 s=5 | CIFAR100 s=10 |
|---|---|---|---|
| | *Acc*@90 | *Acc*@60 | *Acc*@50 |
| FEDAVG | 0.49 | 0.83 | 0.49 |
| PartialFed | 0.14 | 0.56 | 0.38 |
| FedFina | - | - | - |
| FedLP | 2.09 | 0.16 | - |
| FedLPS | - | - | - |
| Flearngene | 1.07 | - | - |
| Ours | **0.06** | **0.08** | **0.24** |

To further validate the low communication cost advantage of the proposed method, we compared its communication cost with FEDAVG in the cumulative training process on the CIAFR100 ($s$=10) dataset, as shown in Figure 3. Compared to the FEDAVG method, which involves interaction with all parameters, our approach achieves a **9.2** × reduction in communication costs. The stabilization of the *learngene* scale around the 40th epoch signifies the acquisition of generalized knowledge among clients and the gradual convergence of the model towards stability. Additionally, Table 6 lists the required communication costs when achieving the target accuracy on different datasets. The proposed method significantly reduces communication overhead compared to FEDAVG, which involves interaction with all parameters. Notably, on the CIFAR100 setting with $s = 10$, the communication cost is nearly halved compared to other methods.

**Effective Initialization Evaluation.** To verify the effectiveness of using small-scale *learngene* fragments for initializing client models in unknown scenarios, we set up 50 clients with untrained class distributions for model initialization and subsequent training. To ensure a fair comparison of the average performance after training, we selected federated model pruning methods capable of capturing partial model information fragments, as listed in Table 3.

Table 3: Performance comparison of federated model pruning methods on *Acc* metric.

| Methods | SVHN | | CIFAR10 | | CIFAR100 | |
|---|---|---|---|---|---|---|
| | $s = 4$ | $s = 5$ | $s = 4$ | $s = 5$ | $s = 10$ | $s = 20$ |
| PartialFed | $91.67 \pm 0.02$ | $91.19 \pm 0.04$ | $63.53 \pm 0.14$ | $61.75 \pm 0.80$ | $\mathbf{53.24} \pm 0.28$ | $\underline{35.43} \pm 0.25$ |
| FedFina | $87.81 \pm 0.06$ | $86.84 \pm 0.08$ | $57.21 \pm 0.11$ | $51.81 \pm 0.03$ | $48.47 \pm 0.14$ | $34.94 \pm 0.12$ |
| FedLP | $90.70 \pm 0.03$ | $88.55 \pm 0.04$ | $\underline{64.61} \pm 0.04$ | $\mathbf{63.24} \pm 0.06$ | $49.30 \pm 0.13$ | $31.51 \pm 0.20$ |
| FedLPS | $79.17 \pm 0.02$ | $77.32 \pm 0.04$ | $52.32 \pm 0.08$ | $45.85 \pm 0.10$ | $40.78 \pm 0.19$ | $31.45 \pm 0.15$ |
| Flearngene | $88.85 \pm 0.01$ | $89.52 \pm 0.02$ | $63.60 \pm 0.09$ | $57.83 \pm 0.07$ | $48.55 \pm 0.10$ | $33.73 \pm 0.11$ |
| Ours | $\mathbf{93.83} \pm 0.02$ | $\mathbf{92.91} \pm 0.03$ | $\mathbf{65.46} \pm 0.10$ | $\underline{63.06} \pm 0.09$ | $\underline{52.44} \pm 0.16$ | $\mathbf{35.49} \pm 0.17$ |

Our method surpasses other approaches in most settings across various datasets, notably achieving an improvement of approximately 2 percentage points over the state-of-the-art PartialFed on the SVHN dataset. While our method performs slightly lower than others in certain CIFAR10 and CIFAR100 settings, these differences are relatively minor and do not significantly impact the overall effectiveness. In the more heterogeneous $s = 20$ setting on CIFAR100, our method consistently outperforms comparable approaches. This demonstrates the efficacy of *learngene* in initializing agnostic client models, exhibiting its scalability and adaptability to unknown and varying class distributions. In addition, Figure 4 shows the test performance curve comparison of our method with advanced model pruning methods on the SVHN with $s = 4$. Although our method initially underperforms, it gradually outperforms the others due to the inherited *learngene*, which requires training to adapt to the new data distribution, leading to a stable performance improvement. This further demonstrates the generalization and scalability of the *learngene* obtained through our method, enabling the initialized models to quickly adapt to agnostic data distributions.

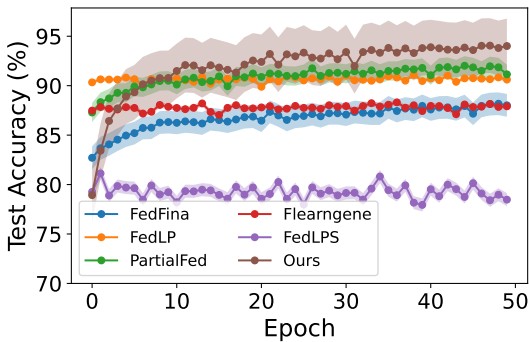

Figure 4: Performance curve comparison.

**Robust Privacy Protection Evaluation.** We conduct Peak Signal to Noise Ratio (PSNR) as a metric to quantify the similarity between original images and those reconstructed by iDLG. A higher PSNR value indicates greater similarity between the images being compared. We integrate differential privacy into the FEDAVG by introducing Gaussian noise with noise levels that $\sigma^2 = 0.001$ to the common gradients. Figure 5 shows a malicious server attack on client data and subsequent image reconstruction using iDLG across different FL methods with different levels of privacy. FEDAVG produces reconstructed images that closely resemble the original, while differential privacy mechanisms show significant improvements. The FLearngene and PartialFed methods upload only a subset of model parameters, providing significant privacy benefits. Note that in the last row of images, FEDAVG has a high PSNR value of 37.58dB, closely resembling the original image, while our method renders them indistinguishable from the original in terms of perceptual similarity. This

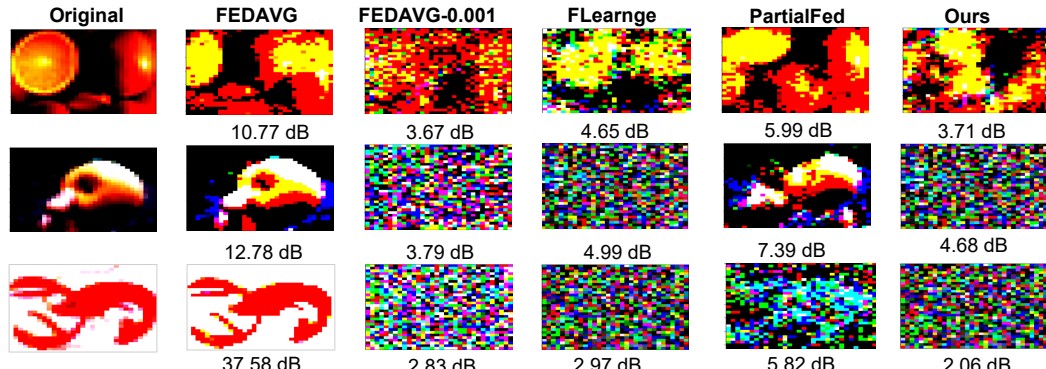

Figure 5: Higher privacy protection. Reconstructing images under iDLG attacks in FEDAVG, FLearngene, PartialFed, and the proposed method. Images are extracted from CIFAR10 and CIFAR100 datasets, with corresponding PSNR reported beneath each recovered image.

further emphasizes that initializing agnostic client models based on the *learngene* downloaded from the server can prevent privacy leakage, even if the server is malicious.

**Ablation Study.** To highlight the contribution of each component or our method to the overall performance, we perform a series of ablation experiments. Our proposed method consists of two integral components. (1) The $\mathcal{L}_{gen}$ to learn the knowledge of others with similar clients and find generalizable *learngenes*. (2) The $\mathcal{L}_{elg}$ to focus on learning elastic *learngenes* to improve the knowledge content of the *learngene*. The results in Table 4 clearly illustrate that both $\mathcal{L}_{elg}$ and $\mathcal{L}_{gen}$ contribute significantly to the performance of the model under various settings. The combined use of both components provides the best results on different datasets, reinforcing the effectiveness of our proposed method.

Table 4: Ablation studies for the proposed method.

| Settings | $\mathcal{L}_{gen}$ | $\mathcal{L}_{elg}$ | SVHN $s = 4$ | CIFAR10 $s = 4$ | CIFAR100 $s = 10$ |
|---|---|---|---|---|---|
| Ours w/o $\mathcal{L}_{gen}$ | ✗ | ✓ | 93.66± 0.03↓0.17 | 64.29± 0.01↓1.17 | 49.36± 0.13↓3.08 |
| Ours w/o $\mathcal{L}_{elg}$ | ✓ | ✗ | 93.55± 0.01↓0.28 | 63.62± 0.06↓1.84 | 48.74± 0.19↓3.70 |
| Ours | ✓ | ✓ | **93.83**± 0.02 | **65.46**± 0.10 | **52.44**± 0.16 |

**Hyperparameter Study.** We exploit different hyperparameters (including $K$ and $\varepsilon$) of proposed method on the CIFAR10 with $s = 4$, where $K$ demonstrates the advantage of multiple global models on the server side and $\varepsilon$ validates how the model retains or changes the elastic *learngene* part with generalization properties as shown in Table 5. For $K$, we observed that multiple global models are more conducive to agnostic clients selecting the optimal initialization model, which verifies the effectiveness of our solution. For $\varepsilon$, our proposed method is relatively stable, which validates the robustness of our proposed solution. We applied these parameters to several different datasets and obtained consistently good performance. In addation, $\lambda_1$ is the hyperparameter controlling the constraint of the local model based on the cluster model, while $\lambda_2$ represents the sensitivity to the strength of the elastic *learngene* constraint. Their analysis is provided in Appendix A.3.

Table 5: Ablation study on various hyperparameters.

| | $K$ | | $\varepsilon$ | | |
|---|---|---|---|---|---|
| | 1 | 4 | 0.1 | 0.5 | 0.9 |
| Acc (%) | 64.16± 0.06 | **65.46**± 0.10 | 65.17± 0.11 | **65.46**± 0.10 | 63.11± 0.06 |

## 4 CONCLUSIONS

In this paper, we delve into the challenges of high communication interaction costs and model initialization for agnostic clients in dynamic agnostic federated learning. We present a Dynamic agnostic Federated Learning with Learngene framework consisting of three modules: Learngene Smooth Learning, Learngene Dynamic Aggregation, and Learngene Initial Agnostic Model, which effectively address the aforementioned challenges. The effectiveness of the proposed approach has been extensively validated on various classification tasks against several popular methods. In the future, we will further investigate how agnostic heterogeneous models can be effectively integrated with Learngene to address initialization and communication issues.

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

# A APPENDIX

We present related work, discussion of proposed framework and detailed experimental setup in the following sections.

## A.1 RELATED WORK

**Federated Learning.** Federated Learning (FL) was first introduced in (McMahan et al., 2017), which demonstrated its effectiveness in learning collaborative models without collecting user data. Compared to centralized learning, FL faces several unique challenges, including non-independently identically distributed and imbalanced data, as well as limited communication bandwidth (Mu et al., 2021; Briggs et al., 2020; Lee et al., 2022; Li et al., 2024). An intuitive approach to reduce communication costs is to quantify weights directly and upload them sparsely (Yi et al., 2024; Jiang et al., 2023; Jiang & Borcea, 2023; Shi et al., 2024b). The FedDrop (Caldas et al., 2018) to reduce the computational burden of local training and the corresponding communication costs in FL. The transferable model design in FedLPS (Jia et al., 2024) uses an adaptive channel model pruning algorithm. Efforts have also been directed towards one-shot FL (Jhunjhunwala et al., 2024; Elmahallawy & Luo, 2023), which aims to achieve satisfactory models with only one round of communication, but requires the shared dataset. Additionally, leveraging class prototypes for low-cost communication has gained attention (Tan et al., 2022). FedTGP (Zhang et al., 2024) proposes using adaptive margin-enhanced contrastive learning to train global prototypes on the server. However, most existing works focus on pruning operations for fixed training clients, with limited attention to dynamic federated learning scenarios in the real world. In contrast, we propose a dynamic agnostic federated learning with Learngene framework, which condenses lightweight *learngene* for participating client model to reduce unnecessary resource consumption in dynamic scenarios.

**Learngene.** The Learngene, as a novel paradigm based on the inheritance principles from biology, enables the condensation of a large-scale ancestral model into *learngene* to adaptively initialize models for various descendant tasks. Wang et al. (Wang et al., 2022) first proposed Learngene based on gradient information from the ancestral model, using limited samples to initialize descendant models. Furthermore, they summarized the three processes of Learngene (Wang et al., 2023): accumulating, condensing, and inheriting. Moreover, Xia et al. (Xia et al., 2024) present the Transformer as a linear extension of Learngene, capable of flexibly generating and initializing Transformers of varying depths. To facilitate the rapid construction of numerous networks with different complexity and performance trade-offs, Shi et al. (Shi et al., 2024a) developed a *learngene* pool method tailored to satisfy low-resource constraints. Simultaneously, (Feng et al., 2024) demonstrated that the transfer of core knowledge through *learngene* can be both sufficient and effective for neural networks. These mentioned approaches underscore the promise of the Learngene paradigm and its feasibility in reducing costs while preserving the essential knowledge of models. **Fisher Information Matrix.** The Fisher Information Matrix (FIM) (Barrett et al., 1995; Ly et al., 2017) is a key concept in statistical estimation theory that encapsulates the information that unknown parameters hold about a random distribution. In deep learning, the FIM has been used to study adversarial attacks (Zhao et al., 2019), guide optimization, and evaluate the information content of parameters (Fasina et al., 2023; Jhunjhunwala et al., 2023; Vallisneri, 2008). For example, (Zhao et al., 2019) utilizes the eigenvalues of FIM derived from a neural network as features and trains an auxiliary classifier to detect adversarial attacks on the eigenvalues. The layer-wise correlation propagation method (Binder et al., 2016) uses the diagonal of FIM to quantify the importance of parameters, thereby improving the interpret ability of the model. The Elastic Weight Removal method (Daheim et al., 2023) weights the individual importance of the parameters via FIM to eliminate hallucinations. These methods all use the diagonal approximation of the FIM to reduce computational complexity and promote a more efficient learning process based on the Fisher information of the parameters.

## A.2 EXPERIMENTAL SETUP

### A.2.1 DATASETS

Our experiments are conducted on the following three real-world datasets: SVHN (Netzer et al., 2011), CIFAR10 (Krizhevsky et al., 2009), and CIFAR100 (Krizhevsky et al., 2009). SVHN is a benchmark digit classification dataset consisting of 600,000 32×32 RGB printed digit images

cropped from Street View house numbers. We select a subset of 33,402 images for training and 13,068 images for testing. The CIFAR10 dataset consists of 60,000 32×32 color images across 10 classes, with 6,000 images per class. It consists of 50,000 training images and 10,000 test images. Similarly, the CIFAR100 dataset contains 100 classes of 600 images each, divided into 500 training images and 100 test images per class.

### A.2.2 SETTINGS

We configure the number of clusters ($K$) to 4, the total number of existing clients ($M$) to 50, and the number of agnostic clients ($N$) to 50. Both seen and unseen classes are equally distributed, each comprising 50% of the total number of classes. We applied K-Means, KNN, and hierarchical clustering algorithms and observed that they exhibited similar performance trends across various FL methods. Therefore, we opted to use the classic K-Means algorithm. Following to the hyperparameter settings in the literature, we set the model pruning probability to 0.5 for the FedLP (Zhu et al., 2023) method and the local model pruning ratio to 0.8 for the FedLPS (Jia et al., 2024) method. We set the threshold $\varepsilon$ to 0.5 for determining the values. The higher $\gamma$ the number of *learngene* layers, the more layers are selected and the better the performance. During the collaborative accumulation training process, we select 10 clients to participate in each training round, and configure 50 agnostic clients for the subsequent Learngene Initial Agnostic Model process. The experiments are conducted on the server equipped with 1 NVIDIA RTX 3090Ti GPU. Each experiment is repeated three times to compute average metrics.

### A.3 ADDITIONAL EXPERIMENTAL RESULTS

**Low-cost Communication Evaluation.** We introduce the another method to simulate heterogeneous scenarios separately by adjusting $\beta$ in Dirichlet distribution (*Dir*). Specifically, we set $\beta = \{0.1, 0.5\}$ for the CIFAR10 basic dataset, as listed in Table 6. It can be observed that in the heterogeneous scenario of the *dir* partitioning, the proposed method still achieves remarkable performance in terms of the *comm* and *cef* metrics, consistent with the results in Table 1. Compared to the state-of-the-art model pruning method, FedLPS, our approach reduces the communication by 0.8 GB. Furthermore, compared to transmitting all parameters using FEDAVG, it achieves a significant reduction of about 11 GB. This demonstrates that the proposed scheme is well-suited for dynamic and agnostic federated learning in practical applications, achieving a better trade-off between low-cost communication and model performance.

Table 6: Comparison with state-of-the-art methods under *Dir* partition strategy.

| Methods | CIFAR10 | | | |
| | $\beta = 0.1$ | | $\beta = 0.5$ | |
| | *Comm* | *Cef* | *Comm* | *Cef* |
|---|---|---|---|---|
| FEDAVG | 15.41 | 0.2303 | 15.41 | 0.2165 |
| PartialFed | 4.32 | 0.0668 | 4.32 | 0.0813 |
| FedFina | 11.38 | 0.1839 | 11.38 | 0.2330 |
| FedLP | 12.58 | 0.1773 | 12.07 | 0.1677 |
| FedLPS | 4.83 | 0.1042 | 4.73 | 0.1866 |
| Flearngene | 6.60 | 0.1061 | 6.61 | 0.1293 |
| ours | 4.03 | **0.0663** | 1.69 | **0.0321** |

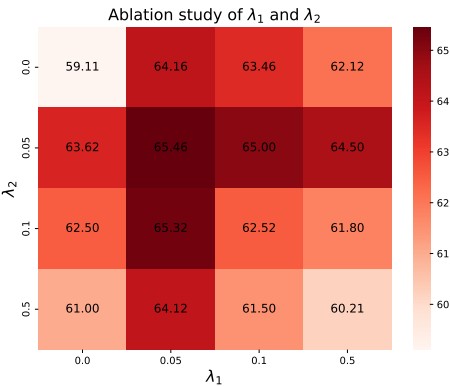

Figure 6: Ablation study on $\lambda_1$ and $\lambda_2$.

**Ablation Study.** We further evaluate the impact of the hyperparameters $\lambda_1$ and $\lambda_2$ on the model performance, as shown in Figure 6. The results show that the combination of $\lambda_1 = 0.05$ and $\lambda_2 = 0.05$ achieves the highest accuracy of 65.46%, indicating this is the optimal setting for the model. Increasing $\lambda_1$ beyond 0.05 leads to a consistent decline in accuracy across all $\lambda_2$ values, while higher $\lambda_2$ values (e.g., 0.5) also result in reduced performance. This analysis highlights the importance of moderate values for both parameters to achieve a balanced trade-off and optimal performance.

