# OpenReview forum: "DFL$^2$G: Dynamic Agnostic Federated Learning with Learngene"
_ICLR.cc/2025/Conference — Submitted to ICLR 2025_

### Official Review · Reviewer_mTsc · 2024-10-24

**Soundness:** 3
**Presentation:** 2
**Contribution:** 3
**Rating:** 5
**Confidence:** 3

**Summary:**

This paper is aiming at addressing two key challenges in Federated Learning (FL):
1) privacy leakage during client-server communication, and
2) communication overhead in transmitting model updates.
To tackle these issues, the authors propose the Learngene framework for Dynamic Agnostic Federated Learning (DAFL). The Learngene framework introduces a mechanism for compressing model updates into learngenes, which capture the most important information while reducing data transmission and mitigating the risk of privacy leakage. Additionally, the framework supports dynamic client participation, allowing clients to join and leave the system flexibly without compromising performance.

**Strengths:**

The paper presents an innovative solution through the introduction of the Learngene framework. By integrating Learngene into the Dynamic Agnostic Federated Learning paradigm, the authors enable efficient model initialization and communication, particularly for agnostic clients that join the system dynamically.

The experimental results are compelling, demonstrating a significant reduction in communication costs while maintaining or even enhancing model accuracy. This highlights the framework's ability to improve both scalability and performance in federated learning environments.

**Weaknesses:**

1) Assume a one-shot dataset in the client. This assumption allows for efficient clustering and model initialization but may limit the framework’s flexibility in handling the common dataset with more samples.
2) Lack of Dynamic Cluster Management: The paper does not address how to manage clusters when they become too large or too small. In cases of high data heterogeneity, more clusters are required to accurately represent the diversity among clients. However, the framework does not discuss mechanisms to dynamically adjust the number of clusters based on client performance, data distribution, or scalability concerns.
3) Insufficient Privacy Guarantees: The paper does not provide strong privacy guarantees. The only implication we have based on your illustration is that "iDLG cannot recover the feature $X \in R^d $ given learngene".
Moreover, the privacy protection is questionable when considering the specifics of the Singular Value Decomposition used in the framework. your $X_i \in R^{1\times d}$, $X_i = U_i \Sigma_i V_i^T$. $U \in R^{1\times1}$, $\Sigma \in R^{1\times d}$ diagonal matrix. Therefore, there are only 2 unknown numbers to recover $X_i$. if ignoring the scale ($U \in R^{1\times1}$), there are only one number left to recover your $X_i$, which would be easy.
Besides, The dimensions are not clearly explained for SVD here. Your $X_i$ should be a matrix $X_i \in R^{1\times d}$


Presentation:
1) Your citation format is incorrect for the entire paper. In latex, most of your citations should be \citep{}. and will be rendered "FL (McMahan et al. 2017)".
2) Since you still have space, I suggest that your algorithm should be placed in the main body of the paper. Because it provides a more general view of how you integrate Learngene smooth learning, learngene dynamic aggregation, and learngene initial agnostic model into one framework.
3) your algorithm line 4. The tilde of $\theta$ is in the wrong place.
4) #276 your mentioned $d=5$. Does this mean that your private data $X \in R^d = R^5 $, If so, is this a typo here?

**Questions:**

How you update the cluster was not specific in algorithm 1. As a new agnostic client join the network, it is added to the nearest cluster as stated in line 18 of Algorithm 1. However, as new clients involve the cluster should be updated. Or is it the cluster only built at the beginning?

---

> ### Author Response · Authors · 2024-11-22
>
> **Q1. Assume a one-shot dataset in the client.**
>
> Thank you for your attention to this detailed question. In our method, the data uploaded to the server for the clustering process consists of one-shot vectors obtained through truncated singular value decomposition of a common dataset. However, the client-specific common dataset remains local and is used exclusively for local model training.
>
> **Q2. Lack of Dynamic Cluster Management.**
>
> Thank you for pointing this out! Firstly, on the server side, clustering is performed based on the low-dimensional left singular vectors uploaded by the client, with the number of clusters can be randomly set based on the data distribution. Subsequently, the unknown clients select the closest clusters to join based on the existing cluster distribution. Therefore, the number of clusters is typically determined initially based on the distribution of the existing clients.
>
> **Q3. Insufficient Privacy Guarantees.**
>
> Thank you for pointing this out! We must emphasize that the vectors uploaded to the server for clustering are used in a one-shot manner and have been shown to be effective in protecting data privacy [1]. Additionally, the iDLG method we employ reconstructs data at pixel-level accuracy based on model gradient information, which is commonly used to evaluate the privacy guarantees of federated learning methods [2-4]. Moreover, during the validation phase, the server recovers the original data solely through the transmitted _learngene_ rather than the uploaded vectors.
>
> **Q4. Presentation**
>
> **1) Citation format；2) Algorithm location.**
>
> Thank you for such valuable and detailed comments. We have revised the citation format throughout the text and moved the algorithm to the main body of the paper.
>
> **3）Describtion of $d$.**
>
> Thank you for your suggestion. Here, $d$ is the number of left singular vectors selected after performing truncated singular value decomposition on the local data. Specifically, we define the decomposition as $\mathcal{X} _ {i,d} = \mathbf{U} _ {i,d} \mathbf{\Sigma} _ {i,d} \mathbf{V} _ {i,d}^T$, where $\mathbf{U} _ {i,d} = [\mathbf{u} _ 1, \mathbf{u} _ 2, \ldots, \mathbf{u} _ d]\in \mathbb{R}^{m \times d}$ (with $d \ll$ rank($\mathcal{X} _ i$)  and $m$ denotes the number of samples for client $i$) represents the top $d$ most significant left singular vectors, capturing the essential features of the underlying data distribution. We follow the [1] and select $d = 5$ to mitigate the risk of data leakage. Additionally, to facilitate linear algebraic computations, we transform the matrix $\mathbf{U} _ {i,d}$ into a vector $\mathbf{u} _ {i,d} \in \mathbb{R}^{md \times 1}$.
>
> **Q5. Description of the cluster updating.**
>
> Thank you for your suggestions. Clusters are established based on the data distribution when the existing clients begin training. When a new client joins the federated learning system, it is assigned to an existing cluster based on similarity and updates the cluster's mean vector (line 20 in Algorithm 1).
>
> ***References:***
> [1] Vahidian S, Morafah M, Wang W, et al. Efficient distribution similarity identification in clustered federated learning via principal angles between client data subspaces[C]//Proceedings of the AAAI conference on artificial intelligence. 2023, 37(8): 10043-10052.
>
> [2] Wu Y, Kang Y, Luo J, et al. Fedcg: Leverage conditional gan for protecting privacy and maintaining competitive performance in federated learning[J]. arXiv preprint arXiv:2111.08211, 2021.
>
> [3]Scheliga D, Mäder P, Seeland M. Combining Variational Modeling with Partial Gradient Perturbation to Prevent Deep Gradient Leakage[J]. arXiv preprint arXiv:2208.04767, 2022.
>
> [4] Ma Y, Yao Y, Xu X. PPIDSG: A Privacy-Preserving Image Distribution Sharing Scheme with GAN in Federated Learning[C]//Proceedings of the AAAI Conference on Artificial Intelligence. 2024, 38(13): 14272-14280.

---

> > ### Comment · Reviewer_mTsc · 2024-11-24
> > **Thank you, I will keep my score.**
> >
> > Thank you for the comments. After reviewing your responses, I have decided to maintain my original score.

---

> > > ### Author Response · Authors · 2024-11-24
> > >
> > > Thank you for your time and effort in reviewing our work and for providing valuable suggestions to help us improve it.

---

### Official Review · Reviewer_CQ3x · 2024-10-31

**Soundness:** 1
**Presentation:** 1
**Contribution:** 2
**Rating:** 3
**Confidence:** 5

**Summary:**

The paper studies dynamic agnostic federated learning, specifically on initializing the client models (by using the learngene paradigm) and achieving better communication overhead while protecting the privacy of the models. They propose DFL$^2$G, which consists of smooth updating, dynamic aggregation, and initial agnostic model.

**Strengths:**

- The paper proposes "collaborating, condensing, initializing" steps analogous to the Learngene paradigm.
- The topic of dynamic agnostic federated learning is important.
- The provided empirical results cover various settings and baseline methods.

**Weaknesses:**

- **Readability**:
   - There are many mistakes, both in the text and notations, creating obstacles for the reader.
   - $\mathcal{X}_{k,i}$: why do you need $k$ here? The local datasets $\mathcal{X}_i$ are not being clustered.
   - Eq.8: why multiplier?
   - [Line 240]: $\sum_{l=1}^L \xi_{k,i}^{(l)} = 1$. How does this sum up to 1? It does not seem to be valid.
   - [Line 229]: Overall, you have the following objective function:
\begin{equation}
\mathcal{L}\_{all} = \lambda \mathcal{L}\_{gen} + \lambda \mathcal{L}\_{elg},
\end{equation}
which gives
\begin{equation}
= \lambda \mathcal{L}\_{cls} (\mathcal{X}\_{k,i}) + \lambda^2 \|\| \theta\_{k,i} - \Theta\_{k} \|\|_2 + \lambda^2 \|\| \theta\_{k,i}^{'} - \Theta_k ||_2,
\end{equation}
and it has issues in the formulation.
   - Typos in lines: 198, 199, 201, 226 (what is the second loss function?), 243 (different subscripts), 272, 283 (why j? you can stick to k.), 313, etc.

- Section 2.4. Problems in the SVD decomposition and formulation. How can you set the data dimension $d$ to 5? $d$ can not equal some other value than its original value.

- Privacy analysis. For a fair comparison with other baseline methods, you need to leverage all available information to reconstruct the samples $\mathcal{X}_i$. Since clients are sharing $V_i$'s with the server, which can aid your reconstruction objective you have (Eq. 12), using the iDLG objective solely is not fair; therefore, it raises a question regarding the results in the paper (Figure 5).

- The number of local epochs is huge (line 335, local epochs = 10), which should not be the case in heterogeneous FL since it makes the clients overfit to their local data.

- The proposal of a new metric. Why propose a metric if you use it only in one table (Table 1)? Also, it is better to see the Acc. measures in Table 1.

- Performance curve comparison (Figure 4). The figure doesn't correspond to what is reported in the table, which questions the study's validity. Also, the proposed method has a high variance (deviation) compared to other methods, which doesn't necessarily mean the method outperforms others. The baseline methods do not improve, having a straight-line performance (FedLP, Flearngene).

- Table captions should be on top.
- Consider citing other works using \citep{}.

**Questions:**

See weaknesses.

---

> ### Author Response · Authors · 2024-11-22
>
> **Q1. Readability**
>
> **1） The description of text and notations.**
>
> Thank you for pointing this out. In the revision, we will simplify the notations of local datasets. We specifically designed $\sum_{l=1}^{L} \xi_{k,i}^{(l)}=1$ to normalize the scores across all layers, ensuring that scores are measured on a unified scale. This allows for clearer comparisons of the relative contributions of each layer to the overall model updates, thereby facilitating the identification of *learngene* within the model.
>
>  **2） The description of objective function.**
>
> Thank you very much for your constructive comments. We will give a more thorough and clearer description of the loss function: $\mathcal{L} _ {all} = \mathcal{L} _ {cls} + \lambda _ 1\mathcal{L} _ {gen} + \lambda _ 2\mathcal{L} _ {elg}$. The corresponding hyperparameter ablation study is shown in Appendix A.3.
>
> **Q2. Problems in the SVD decomposition and formulation.**
>
> Thank you for your valuable feedback! The client applies truncated SVD decomposition to their private data, selecting the top $d$ most significant left singular vectors. These vectors effectively capture the essential characteristics of the underlying data distribution while minimizing privacy leakage [1]. For simplicity in linear algebra computations, the matrix $\boldsymbol{U} _ {i,d} = [\boldsymbol{u} _ 1, \boldsymbol{u} _ 2, \ldots, \boldsymbol{u} _ d]\in \mathbb{R}^{m \times d}$ (with $d \ll rank(\mathcal{X} _ i)$) is further reshaped into a vector form $\boldsymbol{u} _ {i,d} \in \mathbb{R}^{md \times 1}$.
>
> **Q3. Privacy analysis.**
>
> Thanks you for pointing this out! Firstly, the vectors uploaded to the server are used solely for one-shot clustering, and it has been demonstrated in [1] that this approach effectively safeguards data privacy. Second, the iDLG method employed achieves pixel-level accurate data reconstruction based on model gradient information, which is commonly used for privacy verification in federated learning. During the validation phase, we recover the original data solely using the gradient information of the model initialized with the *learngene*, without relying on these vectors. The privacy verification results of different FL methods, as shown in Figure 5, are based on the same experimental setup including the clustering pre-processing between clients.
>
> **Q4. The setting of local epochs.**
>
> Thank you sincerely for your valuable suggestions. Setting local epochs = 10 is a common practice in FL methods [2-4].  Furthermore, using a smaller number of local epochs may help mitigate overfitting to the client data, which could lead to a slight improvement in the performance of our method. We plan to test this in future work.
>
> **Q5. Reason of new metric.**
>
> The reason for introducing the new metric is that, in real-world scenarios, many edge devices face communication constraints. To validate our proposed method's ability to reduce communication costs while maintaining model performance, we also considered a fair comparison with other model pruning methods. The use of this metric in Table 1 is specifically to assess model performance, while other modules are evaluated using different metrics. For example, privacy protection is validated using PSNR.
>
> **Q6. The bias and variance of the experimental results.**
>
> Thank you for pointing this out.  Figure 4 presents the experimental results for SVHN with $s$ = 4, while Table 3 reports the average results over the final 10 epochs, which explains the observed differences. The reason for the high variance lies in the poor performance during the initial epochs. This is because the initialization of the agnostic client models combines *learngenes* with random parameters, requiring an adaptive process to the data. However, the subsequent performance demonstrates a consistent upward trend.
>
> The limited improvement observed in the baseline method indicates that the inherited model possesses strong generalization capabilities, providing good performance from the initial stages. However, this also restricts the ability to learn personalized knowledge for new client, resulting in local models with straight-line performance.
>
> **Q7. Table captions and citation format.**
>
> We will modify these questions in the revised manuscript.
>
> ***References:***
>  [1] Vahidian S, Morafah M, Wang W, et al. Efficient distribution similarity identification in clustered federated learning via principal angles between client data subspaces[C]. AAAI. 2023.
>
>  [2] Wu Y, Kang Y, Luo J, et al. Fedcg: Leverage conditional gan for protecting privacy and maintaining competitive performance in federated learning[J]. arXiv preprint arXiv:2111.08211, 2021.
>
>  [3] Yi L, Wang G, Liu X, et al. FedGH: Heterogeneous federated learning with generalized global header[C]. ACM MM. 2023.
>
>  [4] Wang J, Yang X, Cui S, et al. Towards personalized federated learning via heterogeneous model reassembly[J]. NeurIPS, 2024.

---

> > ### Comment · Reviewer_CQ3x · 2024-11-25
> >
> > Thank you for your response. I appreciate the effort, but I remain unconvinced and recommend another iteration of the work with further clarifications and improvements.

---

### Official Review · Reviewer_yhTj · 2024-11-04

**Soundness:** 3
**Presentation:** 2
**Contribution:** 3
**Rating:** 6
**Confidence:** 4

**Summary:**

The authors introduce  **D2FL**, a novel method designed to address the challenge of initializing local models for agnostic clients in federated learning without necessitating the sharing of a global model. Leveraging the  **Learngene paradigm**, D2FL focuses on the rapid initialization of agnostic models through the use of "learngenes." These learngenes encapsulate essential model knowledge, allowing new or agnostic clients to initialize their local models efficiently by inheriting this distilled information. The primary claims of D2FL include reduced communication overhead and enhanced privacy compared to the standard Federated Averaging (FedAvg) approach. By minimizing the need to transmit large model updates and avoiding the distribution of a global model, D2FL aims to achieve more scalable and privacy-preserving federated learning.

**Strengths:**

1.  **Seemingly Effective Reduction of Communication Costs:**

   D2FL seemingly lowers communication overhead in federated learning  where instead of transmitting full model updates, local updates are compressed into lightweight "learngenes," which are then shared with the server. For a fixed communication budget, the tradeoff is improved. This is shown in experimental work

2.  **Efficient Initialization of Agnostic Client Models:**

 The framework leverages accumulated knowledge from participating clients to generate and store learngenes in a central pool. When new or agnostic clients join the network, they can initialize their models by inheriting these learngenes, facilitating rapid and effective model initialization.

3.  **Improved Privacy Preservation:**

By avoiding the direct sharing of global models and instead using condensed learngenes, D2FL offers improved safety against standard gradient attacks unlike  FedAvg. The authors also highlight that the "privacy" means defense against gradient based attacks only.

**Weaknesses:**

1.  **Ambiguous Notation for Agnostic Clients:**

 The notation used to represent agnostic clients, particularly in lines 128-129, is unclear.

2.  **Scalability Concerns Due to Server-Side Storage Overhead:**

  The server maintains  K cluster models, which introduces significant storage overhead. As the number of clusters increases, the storage requirements may become prohibitive, raising concerns about the scalability of D2FL in large-scale federated learning environments. This limitation is not adequately addressed or acknowledged in the paper. This is especially relevant when comparing with other baselines


3.  **Insufficient Explanation of the Likelihood Function for FIM Computation:**

 The  **Fisher Information Matrix (FIM)**  is utilized within the framework, but the paper does not explicitly explain the likelihood function used to compute it 202-203.

4.  **Complexity of the Learngene Concept:**

 As there are multiple procedures happening in the paper, the introduction and explanation of the Learngene concept are convoluted, making the paper difficult to follow. It required multiple reading to understand some concepts. The authors should simplify the presentation of this concept, possibly by providing more intuitive explanations or systematically develop concepts to improve comprehension.

5.  **Unclear Combined Loss Function:**

  In line 230, the paper presents a combined loss function where the same weight parameter  λ  controls multiple aspects of the loss. The interaction and impact of  λ  on different loss components are not clearly delineated. Also the ablation studies do not incorporate the impact of the hyper parameter adjustment of these seperate learngene and elastic gene loss functins



 6.  **Ambiguities in Experimental Figures and Tables:**

  **Figure 4:**  The dataset and model used in this figure are not clearly specified. Additionally, the performance of D2FL in low epoch regions (e.g., epochs less than 10) is smaller than some baselines other methods that perform better under these conditions.  This needs to be acknowledged.

   **Table 4:**  The table does not include standard deviations. Furthermore, it fails to separately evaluate the impact of elasticity and the Learngene component, despite elasticity being a core component of the paper. Same hyper parameter controls both the loss function so it is difficult to establish the impact of these seperate loss functions. This omission makes it challenging to determine the individual contributions of each component to the overall performance.

  **Table 5:**  Similar to Table 4, Table 5 lacks descriptive information about the datasets used and the statistical measures reported.

7.  **Absence of Theoretical Convergence Guarantees:**

    The paper does not provide any theoretical analysis or proofs to support the convergence of the Learngene-based initialization method.

**Questions:**

Please refer to weaknesses

---

> ### Author Response · Authors · 2024-11-22
>
> **Q1. Ambiguous Notation for Agnostic Clients.**
>
> Thank you for your suggestion. We have provided a clear explanation of this in the revised manuscript, specifically in lines 125-130.
>
> **Q2.  Scalability Concerns Due to Server-Side Storage Overhead.**
>
> First, the bottlenecks of federated learning are typically centered around communication overhead [1, 2] and the limited storage capacity of edge devices [3]. Thus, the goal of our proposed method is to reduce costs by leveraging communication based on *learngenes*.  Second, the storage overhead at the server side is generally caused by high-frequency data generated by time-series sensors [4]. In contrast, in our approach, the uploaded vectors used for clustering are one-shot, and the number of cluster models can be customized.
>
> **Q3. Insufficient Explanation of the FIM Computation.**
>
> Thanks. We have described the FIM computation in more detail in lines 208-213 of the manuscript.
> We approximate the diagonal of the Fisher information matrix for each parameter indexed by $j$ in the model $\tilde{\theta} _ {i}$ (refers to $\tilde{\theta} _ {k,i}$), expressed as $F _ {i,j}=\mathbb{E}\left[\left(\frac{\partial \log \mathcal{h}( \tilde{\theta} _ i \mid \mathcal{D} _ i)}{\partial \tilde{\theta} _ {i,j}}\right)^{2}\right]$. Here, the likelihood function $\mathcal{h}( \tilde{\theta} _ i \mid \mathcal{D} _ i)$ represents the fitness of the model parameters given the data $\mathcal{D}_i$. In our implementation, the log-likelihood is indirectly computed using the log_softmax output, which corresponds to the log probability of the correct class label. The Fisher information diagonal is then obtained by computing the gradient of the log-likelihood with respect to the model parameters, aligning with the concept of Fisher information.
>
> **Q4. Complexity of the Learngene Concept**
>
> Thank you for pointing this out. Learngene is an innovative paradigm inspired by biological genetic evolution, designed to distill inheritable knowledge from ancestral models in an open-world setting, creating *learngene* that enable efficient adaptation to new tasks.
> We deeply consider the "**Collaborating & Condensing & Initializing**" mechanism in dynamic FL based on the perspective of "*Accumulating & Condensing & Inheriting*" in Learngene to enhance model interpretability. Specifically, this involves leveraging "collaborative learning of local models, condensation of generalized knowledge, and initialization of agnostic client models"  to achieve low-cost communication and adapt to dynamic agnostic scenario.
>
>  **Q5. Unclear Combined Loss Function.**
>
> Thank you very much for your constructive comments. We set different hyperparameters $\lambda_1$ and $\lambda_2$ for different loss functions, and performed ablation studies in Appendix A.3.
>
> **Q6. Ambiguities in Experimental Figures and Tables.**
>
> Thanks. We acknowledge that the performance during the first 10 rounds is lower than that of other methods. The reason for this is that the *learngenes* used to initialize the agnostic client models are information fragments from the model, while the other components are initialized randomly. As a result, the model requires an adaptation process to adjust to the new task. However, in the subsequent rounds, the performance consistently outperforms that of other methods, demonstrating the model's generalization capability and providing a solid foundation for further training.
> **Table 4 and Table 5:** We have added detailed descriptions of the datasets and specific statistical measures to Tables 4 and 5. While the original manuscript used the same hyperparameter settings, we also conducted ablation studies on the Elastic and Learngene components, as shown in the second row of Table 4.
>
>  **Q7. Absence of Theoretical Convergence Guarantees.**
>
> Thanks. First, we must emphasize that the proposed dynamic agnostic federated learning addresses a practically significant application problem, aiming to overcome the limitations of traditional federated learning in dynamic and agnostic scenarios. Second, we introduced the innovative concept of "**Learngene**," which provides a highly practical solution. Finally, to validate the effectiveness of the proposed method, we conducted a series of experiments to evaluate the model's performance in terms of training, communication efficiency, and privacy preservation.
>
> ***References:***
> [1] Luping W., Wei W., Bo L. I. CMFL: Mitigating communication overhead for federated learning. ICDCS, 2019.
> [2] Malaviya S., Shukla M., Lodha S. Reducing communication overhead in federated learning for pre-trained language models using parameter-efficient finetuning. PMLR, 2023.
> [3] Dai Y., Xu D., Maharjan S., et al. Joint load balancing and offloading in vehicular edge computing and networks. *IEEE Internet of Things Journal*, 2018.
> [4] Zhang T., He C., Ma T., et al. Federated learning for internet of things. *ACM Conference on Embedded Networked Sensor Systems*, 2021.

---

> ### Comment · Reviewer_yhTj · 2024-11-23
>
> Thank you for this.My main concerns centered around three areas:
> 1) Presentation and writing of the paper ie: systematic development of the concept learn gene
> 2) Issues in the ablation part of the paper ie: Combined Loss function
>  3) Lack of theoretical convergence guarantees.
>
> The authors have addressed 2) for me. 1) can be addressed in a separate version. 3) is not attended at the moment by the authors.
>
> Therefore, I am raising the score slightly.
>
> Thanks

---

> > ### Author Response · Authors · 2024-11-23
> >
> > Thank you very much for taking the time to review our paper and raise the score!
> >
> > The theoretical proof requires a high degree of abstraction regarding the Learngene's form and a series of assumptions about the learning problem's setup. We believe this is an important issue to address for the Learngene to evolve into a *provably effective framework* with practical promising in the future. Prior to this, our work has actively explored the adaptability of this framework and dynamic agnostic federated learning, which we believe is still valuable. Once again, we express our sincere gratitude!

---

### Official Review · Reviewer_M8kk · 2024-11-04

**Soundness:** 2
**Presentation:** 2
**Contribution:** 2
**Rating:** 3
**Confidence:** 4

**Summary:**

This manuscript proposes a framework, called DFL2G, to address two main challenges in federated learning: (1) initialization of the client model parameters for new "agnostic" clients and (2) to reduce communication overhead between clients and server during training process. The framework consists of three modules: Learngene Smooth Learning, Learngene Dynamic Aggregation, and Learngene Initial Agnostic Model, to effectively address these challenges. Experimental results demonstrate that the approach effectively reduces communication cost while maintaining comparative classification accuracy.

**Strengths:**

1. This paper proposes an innovative approach for federated learning, which dynamically initializes effective parameters for new clients and utilizes  Learngene concept to reduce communication overhead and strengthen privacy.
2. The results show that the performance of the proposed method is comparable with the baselines.
3. The paper is well-structured.

**Weaknesses:**

1. Lack of convergence proof and theoretical support.
2. The experimental results are limited. Further the authors have not considered different heterogeneous settings in their experiments.
3. There is no comparison with the baselines having similar objectives (e.g., FedProto, FedTGP).

**Questions:**

1. I believe that the "cef" measure in Table 1 doesn't provide a fair comparison, as there is no direct relation between communication cost and accuracy.
2. It would be nice to see more experimental support, including diverse datasets and non-IID scenarios with different data heterogeneity levels (α = 0.05, 0.5, 0.1).
3. Also the authors should consider to include one or two standard FL baseline like SCAFFOLD, FedProto, FedTGP, to better demonstrate method's superiority.

---

> ### Author Response · Authors · 2024-11-22
>
> **Q1. Lack of convergence proof and theoretical support.**
>
> Thank you for your suggestions. First, we must emphasize that the proposed dynamic agnostic federated learning addresses a practically significant application problem, aiming to overcome the limitations of traditional federated learning in dynamic and agnostic scenarios. Second, we introduced the innovative concept of "**Learngene**," which provides a highly practical solution. Finally, to validate the effectiveness of the proposed method, we conducted a series of experiments to evaluate the model's performance in terms of training, communication efficiency, and privacy preservation.
>
> **Q2.	 The experimental results are limited.**
>
> We really appreciate your constructive feedback! Due to time constraints, we added another non-IID scenario ($\beta$ = 0.5, 0.1) using the Dirichlet distribution on the benchmark dataset CIFAR10, as shown below:
> | Methods    | $\beta$ = 0.1|        | $\beta$ = 0.5   |     |
> |------------|------------|------------|------------|------------|
> |            | *comm*       | *cef*        | *comm*       | *cef*        |
> | FEDAVG     | 15.41      | 0.2303     | 15.41      | 0.2165     |
> | PartialFed | 4.32       | 0.0668     | 4.32       | 0.0813     |
> | FedFina    | 11.38      | 0.1839     | 11.38      | 0.2330     |
> | FedLP      | 12.58      | 0.1773     | 12.07      | 0.1677     |
> | FedLPS     | 4.83       | 0.1042     | 4.83       | 0.1866     |
> | Flearngene | 6.60       | 0.1061     | 6.61       | 0.1293     |
> | **ours**       | 4.03      | **0.0663**     | 1.69       | **0.0321**    |
>
> The table below presents a performance comparison of model training after initialization for agnostic clients:
>
> |            Methods      | $\beta$ = 0.1          | $\beta$ = 0.5        |
> |------------------------|---------------|--------------|
> | PartialFed         | 59.62         | 51.29        |
> | FedFina            | 58.10         | 47.61        |
> | FedLP              | 60.13         | 51.69        |
> | FedLPS             | 59.33         | 51.96        |
> | Flearngene         | 59.06         | 51.32        |
> | **Ours**           | **61.14**     | **52.26**    |
>
>
> **Q3. There is no comparison with the baselines having similar objectives.**
> Thank you very much for your constructive comments. While it might appear that our approach shares similar objectives with FedProto[1] and FedTGP[2], there are fundamental distinctions in scope and purpose. FedProto and FedTGP aim to address joint optimization in distributed networks, leveraging class prototypes to minimize communication costs. In contrast, our method is not limited to low-cost communication but is designed to ensure effective initialization for models on agnostic clients.
>
> In the dynamic agnostic FL scenario, agnostic clients and previously trained clients share no overlapping classes ( $\mathcal{C} _ {\text{agnostic}} \cap \mathcal{C} _ {\text{known}} = \emptyset$). Methods like FedProto and FedTGP rely on server-learned global class prototypes, which is unsuitable for our scenario. Using old class prototypes to initialize distinct new class prototypes on agnostic clients is infeasible and unjustifiable due to the lack of overlap between their class distributions. We will discuss prototype-based communication approach in the revised manuscript.
>
> **Q4. Questions about the *cef* measure.**
>
> Thank you for your insightful question. In FL, discussions often focus on overcoming communication constraints and improving model performance during collaborative learning. Our proposed "cef" measure is specifically designed for scenarios where communication resources are limited, but devices need to learn from the knowledge of others to improve their own performance. Since there is a trade-off the communication costs and model performance, the "cef" measure was introduced to allow fair comparisons with methods that require the transmission of entire models.
>
> ***References:***
> [1] Tan Y, Long G, Liu L, et al. Fedproto: Federated prototype learning across heterogeneous clients[C]//Proceedings of the AAAI Conference on Artificial Intelligence. 2022, 36(8): 8432-8440.
> [2] Zhang J, Liu Y, Hua Y, et al. Fedtgp: Trainable global prototypes with adaptive-margin-enhanced contrastive learning for data and model heterogeneity in federated learning[C]//Proceedings of the AAAI Conference on Artificial Intelligence. 2024, 38(15): 16768-16776.

---

> > ### Comment · Reviewer_M8kk · 2024-11-27
> > **No change in the score**
> >
> > Thank you for the comments. After reviewing your responses, I have decided to maintain same score.

---

### Meta-Review · Area_Chair_xSvs · 2024-12-17

**Metareview:**

This paper proposes $DFL^2G$, an initialization technique for dynamic agnostic federated learning. The challenge that the paper overcomes is that for securely and effectively initializing models for agnostic clients. The authors are inspired by the recently proposed Learngene paradigm, which involves compressing a large ancestral model into meta-information pieces that can initialize various descendent task models.

The authors primarily justify the use of their method experimentally, showing the effectiveness of their method in achieving low-cost communication, robust privacy guarantees and effective initialization for agnostic clients.

The paper suffers from several deficiencies, which make it not suitable for publication in the current form. Several reviewers have the same concerns, which I'll reiterate here.

i) Lack of theoretical convergence guarantees;
ii) Lack of privacy guarantees;
iii) Poor exposition/notation, leading to poor comprehension by readers;
iv) Lack of comparisons with the baselines having similar objectives.

The main issue, to me, for a paper on optimization for federated learning is (i). Hence, I have to recommend rejection.

**Additional Comments On Reviewer Discussion:**

The reviewers and authors had a robust, healthy and respectful exchange, which is nice to see. However, at the end of the day, the reviewers were insistent on the deficiencies I outlined above. I agree that this paper is not mature enough, at this point in time, to be published. I believe that the authors are cognizant of these weaknesses and will take the reviewers' comments into consideration in the next version of this paper.

---

### Decision · Program_Chairs · 2025-01-22

Reject